# FAST LINEAR INTERPOLATION FOR PIECEWISE-LINEAR FUNCTIONS, GAMS, AND DEEP LATTICE NETWORKS

## ABSTRACT

We present fast implementations of linear interpolation operators for piecewise linear functions and multi-dimensional look-up tables. We use a compiler-based solution using MLIR to accelerate this family of workloads. On two-layer and deep lattice networks, we show these strategies run $5 - 10\times$ faster on a standard CPU compared to a C++ interpreter implementation that uses prior techniques, producing runtimes that are 1000s of times faster than TensorFlow 2.0 for single evaluations.

## 1 INTRODUCTION

Linearly-interpolated look-up tables (LUTs) are a core operation of many machine-learned models, ranging from simple one-dimensional piecewise-linear (PWL) functions (see Fig. 1), to $D$-dimensional generalized additive models (Hastie & Tibshirani, 1986) (GAMs) that are a linear combination of a PWL on each input variable, to $D$-dimensional interpolated LUTs also known as *lattices*. Interpolated LUTs can be used to efficiently express arbitrarily flexible functions over an arbitrary number of inputs $D$ by increasing the size of the underlying LUTs, or by using ensembles or multiple layers of lattices (Canini et al., 2016; You et al., 2017; Cotter et al., 2019).

Interpolated LUTs have long been a standard choice for applications where fast evaluation and flexible models are needed. For example, two-layer models formed by PWLs followed by lattices are part of the International Color Consortium standard for colorspace transformations (Sharma & Bala, 2002). Garcia & Gupta (2009) showed that lattices can be trained by empirical risk minimization. As concerns grow about the black-box nature of AI, the highly-structured nature of interpolated LUTs has made them a popular choice for producing AI with safety guarantees such as bounded outputs and satisfying *shape constraints* (Groeneboom & Jongbloed, 2014; Chetverikov et al., 2018), such as *monotonicity* which guarantee that a model only responds positively to increases in designated inputs (Barlow et al., 1972; Howard & Jebara, 2007; Gupta et al., 2016; Canini et al., 2016; You et al., 2017), and other shape constraints (Pya & Wood, 2015; Chen & Samworth, 2016; Gupta et al., 2018; Cotter et al., 2019).

Fast evaluation of models is paramount for real-time estimation needs, and cheap evaluation is key to lowering the costs of AI. In theory, what makes interpolating LUTs so fast is that only a small subset of parameters is needed for any given evaluation, in contrast to models like DNNs where every parameter may be needed for every evaluation.

In this paper we investigate how fast one can evaluate interpolated LUTs on standard CPUs. The main contributions of this paper are: *(i)* we give two new strategies to reduce the runtime of PWLs, and *(ii)* we show how to optimize the kernels for two types of multi-dimensional linear interpolation (simplex and multilinear). Overall we show a 5-10x speed-up for the runtime of real-world lattice models, compared to a C++ interpreter implementation that uses prior techniques, producing runtimes that are 1000s of times faster than TensorFlow 2.0 for single evaluations. All comparisons are on the same standard CPUs.

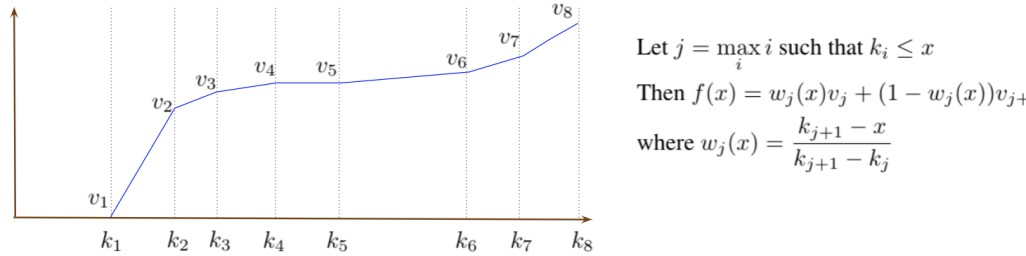

Figure 1: An example PWL defined by 8 key-value pairs, and the formula for linearly interpolating the PWL at any point $x \in [k_1, k_8]$ (inputs outside the domain are clipped to the domain).

## 2 BACKGROUND ON LINEAR INTERPOLATION OPERATORS AND SOME INITIAL ANALYSIS

We review PWLs and the two standard types of multi-dimensional interpolation, and what the challenges are in making them run fast.

### 2.1 PIECEWISE-LINEAR FUNCTIONS (PWLs)

PWLs are an old technique for efficiently representing functions (Sang, 1875; Perry, 1899; Farr, 1860). Multi-layer machine-learned models may use thousands of PWLs to transform the inputs in their first-layer, which makes the overall performance of a lattice model highly sensitive to the runtime of the PWL kernel. As illustrated in Fig. 1, we define a PWL by N key-value pairs $(k_i, v_i)_{i=1}^N$, where the keys are sorted $k_i < k_{i+1}$, and the values are interpolated to evaluate any input $x \in [k_1, k_N]$ as per the formula in Fig. 1.

The key challenge to efficiently evaluating PWLs is to quickly find the index $j$ in the linear interpolation formula in Fig. 1 needed for a given $x$. We estimate the cost of a memory load and compare at 2 cycles if speculative memory loads are allowed, and 4 cycles if they are not, due to the 2-3 cycle latency of a memory load. Each branch misprediction costs an expected 15-20 cycles depending on architecture (Fog, 2018). As a result, we estimate the overall expected cost to be $E[\# \text{ Cycles}] = 4 \times E[\# \text{ Comparisons}] + 17 \times E[\# \text{ Mispredictions}]$. Here, we assume that all parameters are within the L1-cache because there are few enough parameters to be loaded together.

#### 2.1.1 LINEAR SEARCH FOR PWLs IS SLOW, EVEN FOR SMALL PWLs

Linear search over the $N$ keypoints requires $\mathbb{E}[\# \text{ Comparisons}] = \frac{N+1}{2}$. However, as we show in this section, it is the branch mispredictions that are the bigger problem. Recent recommendations for machine-learned lattice models, e.g. (Gupta et al., 2016; 2018), are to set the keypoints based on the quantiles of the training examples for that input: assign keypoint $k_1$ to the minimum possible value of the PWL's domain, assign the last keypoint $k_N$ to the maximum possible value of the PWL's domain, and assign the remaining $N-2$ keypoints to equally-spaced quantiles of the training examples' inputs. We assume the keypoints stay fixed at these values and are not trained (though their corresponding values $\{v_i\}$ are trained). Quantile keypoints are good for machine-learning because each keypoint sees roughly $1/N$ of the training examples, reducing the chance of overfitting any of the trained PWL values $\{v_i\}$. Quantile keypoints also aid in interpretability because the PWLs can be visualized, and quantile keypoints show how the training data was distributed over the input domain. However, we next show that quantile keypoints are terrible for PWL runtimes because each keypoint is equally likely to be the left-keypoint for a random input $x$.

Recall that a *branch predictor* predicts whether or not a given branch is taken, and an *optimal branch predictor* always predicts the outcome associated with the highest probability. During the linear search, a branch prediction will predict whether the for-loop over the keypoints will stop, for each $i = 1, \ldots, N$. Typically, a branch predictor is able to access a summary of its history, and any static information the compiler may be able to provide. For quantile keypoints, each of the

first $N - 1$ keypoints is equally likely to be the correct index. Thus the optimal branch prediction is to continue unless the linear search has reached $i = N - 2$, in which case there is a 50-50 chance of either of the remaining two keypoints being the right one. However, $(N - 2)/(N - 1)$ of the examples will find their correct index before the linear search reaches $i = N - 2$, which means the branch prediction will be wrong once with probability $(N - 2)/(N - 1)$, producing $E[\# \text{ Mispredictions}] = (N - 2)/(N - 1)$.

### 2.1.2 BINARY SEARCH FOR PWLS

In a branch-free implementation of binary search with known depth, the compiler is able to fully unroll the structure and thus avoid branch mispredictions. Additionally, a well-optimized binary search implementation is able to perform each step of the binary search in approximately 6 cycles (Khuong, 2012). It thus takes $6 \times \lceil \log_2(N) \rceil$ cycles to find the appropriate location. This makes binary search roughly $2\times$ more efficient than linear search even for $N = 3$ to $10$, and is the baseline that any proposed indexing must beat.

### 2.1.3 A MAP-TO-INDEX FUNCTION FOR PWLS

More abstractly, the goal is to construct an efficient function that can map an input $x$ to the correct index $j$. An old trick is to build an auxiliary LUT over $[k_1, k_N]$ with $B$ uniformly-spaced buckets, use that to map $x$ to a bucket, and then linearly-search through all the keypoints that fell in that bucket. However, with irregularly-spaced keypoints, a uniform bucket can still have $O(N)$ keypoints to search through. Aus & Korn (1969) proposed constructing a hierarchy of such auxiliary *uniformly*-spaced LUTs to better cover irregular keypoints. O'Grady & Young (1991) proposed using a sufficiently large $B$ such that no uniform bucket contains more than one keypoint, but at the cost of potentially large $B$. An analogous problem arises in database indexing, where recent work has proposed machine-learning a two-layer DNN to produce the map-to-index function (Kraska et al., 2018). Our proposed solution will be in a similar spirit but lighter-weight.

## 2.2 MULTILINEAR INTERPOLATION

Consider one cell of a $D$-dimensional LUT, which without loss of generality is a $D$-dimensional unit hypercube parameterized by LUT values $v \in \mathbb{R}^{2^D}$ corresponding to the $O(2^D)$ vertices of the hypercube. For an input $x \in [0, 1]^D$, *multilinear interpolation* outputs $f(x) = \sum_{i=1}^{2^D} v_i w_i(x)$, where $v_i$ is the stored multi-d LUT value for the $i$th vertex in the $D$-dimensional unit hypercube, and $w_i(x)$ is the multilinear interpolation weight on the $i$th unit hypercube vertex $\xi_i \in [0, 1]^D$ taken in lexicographical order, computed from $x \in [0, 1]^D$ as:

$$w_i(x) = \prod_{d=1}^{D} x[d]^{\xi_i[d]} (1 - x_d)^{1 - \xi_d} \text{ for all } i = 1, \dots, 2^D \tag{1}$$

For example, for $D = 1$, $\xi_1 = 0$, $\xi_2 = 1$, and thus $w_1(x) = 1 - x$ and $w_2(x) = x$, the same as in the PWL formula in Fig. 1 given that there are only two keypoints $k_1 = 0$ and $k_2 = 1$. For $D = 2$, this is standard bilinear interpolation.

Gupta et al. (2016) gave an $O(2^D)$ dynamic programming algorithm for computing (1). We note that while asymptotically efficient, the dynamic programming solution introduces a loop-carry dependency, and thus prevents critical compiler optimizations, a problem we show how to avoid.

## 2.3 SIMPLEX INTERPOLATION

Simplex interpolation is a more efficient linear interpolation of a $D$-dimensional LUT cell that produces a locally linear surface comprised of $D!$ pieces. For each input $x \in [0, 1]^D$, the $D$ components of $x$ are sorted, the resulting sort order determines a set of $D + 1$ vertices whose simplex is guaranteed to contain $x$, and then a sparse inner product is taken with the corresponding $D + 1$ LUT values to produce $f(x)$ (Rovatti et al., 1998; Weiser & Zarantonello, 1988; Gupta et al., 2016). This algorithm is the same as the Lovász extension in submodularity (Bach, 2013).

Gupta et al. (2016) gave runtimes for single-layer models that only did either multilinear and simplex interpolation and were implemented in C++ on a single-threaded 3.5GHz Intel Ivy Bridge proces-

sor: for $D = 4$ inputs both interpolations ran in about 50 nanoseconds, but for $D = 20$, simplex interpolation ran in 750 nanoseconds, and multilinear interpolation ran in 12 milliseconds, around $15,000\times$ slower. Despite the slower runtime for $D > 4$, multilinear interpolation might be preferred because it produces a smoother surface and might produce more accurate models for some problems. Furthermore, the large-scale multiplications needed for multilinear are a better match than the sorting needed for simplex for machine learning libraries such as TensorFlow.

Like with PWLs, branch prediction poses a significant challenge when implementing the simplex interpolation kernel. For example, we found that LLVM defaults to using either a hard-coded insertion sort or quicksort depending on the input size, which is determined at runtime. However, we note that this run-time decision is not needed for machine learning models, because the number of inputs $D$ is fixed and known. Using `std::sort<std::pair<double, int>>`, we found that the sorting operation accounts for approximately $70\%$ of the simplex kernel's overall runtime.

## 3 COMPILATION MATTERS DUE TO LARGE DISPATCH OVERHEAD

Recent work investigated the use of compilers to speed-up machine-learned models (Rotem et al., 2018; Chen et al., 2018; Vasilache et al., 2018), but focused on models with *large operations*, such as convolutional networks or large matrix-multiplies models, where the operations takes hundreds of times longer than the dispatch process. For example, ResNet-34 performs 3.6 billion floating point ops across 34 layers, averaging over 100 million floating point operations per kernel (He et al., 2016). In contrast, for *small operation* models like linear interpolation, the cost of dispatch can be bigger than the computation, and thus reducing dispatch overhead becomes key. For example, the proprietary multi-layer lattice models described in this paper execute at most a few thousand floating point ops per kernel, and many useful models use a few hundred or fewer ops per kernel. Additionally, many usages of lattice models are to evaluate a single example at a time in a latency-sensitive pipeline, negating the chance to amortize overhead by batching.

To reduce dispatch overhead, we use the MLIR framework (Google, 2019) to convert the trained models into compiler-optimized C++ code. This change alone removes a significant portion of the dispatch overhead, providing a speedup of $2 - 3\times$. In the next sections, we show how to reduce the runtime further by taking advantage of the details of the linear interpolation ops.

## 4 HOW TO MAKE PWLS RUN FASTER

We describe two complementary techniques for making PWLs run faster. In Section 4.1 we show how to construct a better auxiliary index-mapping function that takes into account the spacing of the keypoints. In Section 4.2 we show that ensemble or deep lattice models often pass the same input through multiple PWLs, allowing us to remove redundant index searches.

### 4.1 KEYPOINT DEPENDENT OPTIMIZATION

We propose a new way to construct an index mapping function $m$ that first transforms the keypoints to be more uniformly-spaced, and then applies an auxiliary LUT to map the transformed space to an index using an optimal number of uniform buckets. The resulting implementation is constant-time in the number of pieces in the PWL.

Let $C_m(x)$ be the cost of evaluating the mapping $m$ on an input $x$, and $C_{m,\text{Adjust}}(x)$ be the cost of correcting the predicted keypoint index. This naturally produces the procedure $x \to m(x) \to \text{Adjust}(m(x))$. If we know the model will be evaluated on random examples $x \sim P$, then we propose finding the mapping $m^*$ from a family of possible mapping function $\mathcal{M}$ that minimizes the expected cost over random examples to be evaluated:

$$\underset{m \in \mathcal{M}}{\arg\min} \, \mathbb{E}_{x \sim P}[C_m(x) + C_{m,\text{Adjust}}(x)]. \tag{2}$$

We propose using mappings of the form $m(x) = \text{LUT}[\lfloor \alpha + \beta T(x) \rfloor]$ where $T : \text{supp}(P) \to \mathbb{R}$ is some 1-d monotonic transform, and LUT has $B$ uniformly spaced buckets of size $\beta$ that each map to the smallest index encountered in the bucket's interval.

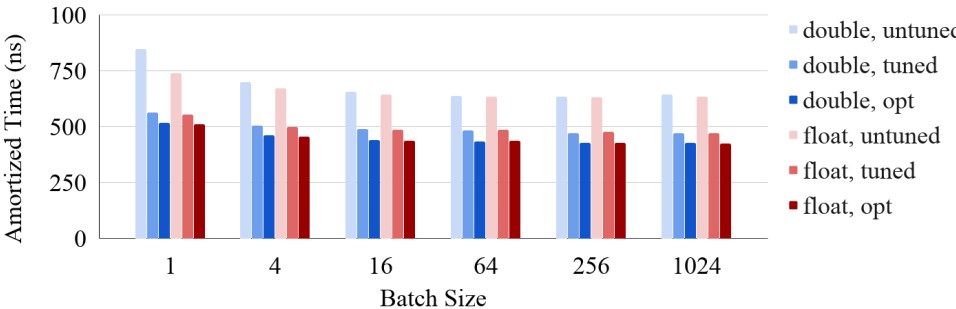

Figure 2: The effect of the proposed optimized index-mapping on a calibrated multilinear model with 8 PWLs. *Untuned* refers to the compiled implementation using an auxiliary mapping function that has $B = 50$ uniform buckets. *Tuned* refers to our proposed keypoint-dependent tuned index mapping. *Opt* also uses a bounded linear search

We estimate the cost $C_m(x)$ to be $C_T(x) + \lambda B + 4$, where $\lambda$ is a relatively insensitive parameter that should represents the cache behavior cost of large arrays, and is set to $0.001$ for all of our experiments.

For most modern CPUs, the branch-misprediction penalty is high, so we propose Adjust and $T$ functions that are branch-free. A consequence is that the cost $C_T(x) = C_T$ is independent of $x$, and we set the Adjust function to be a fixed-step linear search that always executes the worst-case number of steps needed. With these choices, the mapping optimization of (2) becomes:

$$\underset{T \in \mathbb{T}, \alpha \in \mathbb{R}, \beta \in \mathbb{R}, B}{\arg\min} \left( C_T + \lambda B + \left( \max_{x \in \text{supp}(P)} \text{Index}(x) - \text{LUT}[\lfloor \alpha + \beta T(x) \rfloor] \right) \right), \tag{3}$$

where $\text{Index}(x)$ is the correct left keypoint index for $x$, and $\text{supp}(P)$ is the support of the probability distribution of $x$. Note the adjustment cost $\max_{x \in \text{supp}(P)} \text{Index}(x) - \text{LUT}[\lfloor \alpha + \beta T(x) \rfloor]$ is created by the pair of points with maximally different indices that get mapped to the same bucket:

$$\max_{x, z \in \text{supp}(P)} \text{Index}(x) - \text{Index}(z) \text{ such that } \lfloor \alpha + \beta T(x) \rfloor = \lfloor \alpha + \beta T(z) \rfloor \tag{4}$$

Next we show how to take advantage of the case that the keypoints are *quantile keypoints* as described in Section 2. If we set the LUT to span the keypoint range $[k_1, k_N]$, the worst case adjustment needed in (4) is defined by the maximum number of the $N - 2$ quantiles (each of which corresponds to a keypoint, and hence a possible Index) that end up in the same bucket. Thus, the optimal transform $T$ would map the irregularly-spaced keypoints to a uniform spacing, and the resulting worst-case adjustment would be $\lceil (N - 2)/B \rceil$. Actually finding that optimal $T$ we leave as future work. Experimentally, we approximated $T$ by choosing the best out of a small fixed set of simple monotonic transformations, including a fast approximate $log_2(x)$ and an approximate $2^x$, which resulted in needing at most 3 steps for the Adjust linear scan for our models. We note that the problem of constructing $T$, or $m$ in general, may be a problem well suited for superoptimizers.

Fig. 2 gives example speed-ups for a two-layer calibrated lattice model with 8 PWLs in its first layer (described in Section 7 as *whole path model*).

## 4.2 EFFICIENT HANDLING OF SHARED INDEX PWLS

Next, we consider models that have *shared index PWLs* such that the same input is passed through multiple PWLs in order to transform the same input in different ways. As a very simple example, the one-dimensional function $f(x) = 3\log(x) + 4\sqrt{x} + 6x + 2x^2$ for $x \in [0, 1]$ can be represented as the sum of four PWLs on the same input. This situation commonly arises with ensembles of calibrated lattices (Canini et al., 2016). If the keypoints are the same for each PWL on an input (as is the case for *quantile keypoints*), then the work to determine the proper parameter index is duplicated across all PWLs that act on the same input. We propose a model transformation step to convert the PWL execution into separate indexing and interpolation phases so we can map each input $x$ to its index $j$ *only once* for all PWLs on the input $x$.

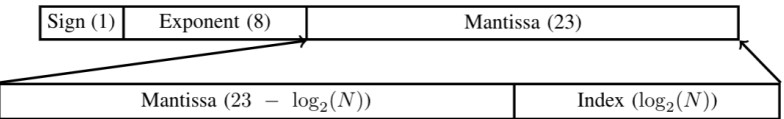

Figure 3: We propose encoding the index in the low-order bits of the interpolation weight for better memory usage. For the double-precision floating point format the exponent instead has 11 bits and the mantissa has 53 bits.

The proposed indexing phase takes in an input $x$, and finds the index $j$ and the interpolation weight $w_j(x)$ to form the pair $\langle j, w_j(x) \rangle$, which is needed by all the PWLs that act on $x$. Then the proposed interpolation phase consumes this pair to compute each PWL's interpolated value using its PWL values as $v_j w_j(x) + v_{j+1}(1 - w_j(x))$, as in Fig. 1.

However, this requires handling the pair $\langle j, w_j(x) \rangle$ efficiently. A naive solution would encode this information literally as a pair of values $\langle j : Int, w_j(x) : Real \rangle$. To avoid the need to separately track the integer portion of the pair, we propose a bit-packing technique that encodes the index in the low-order bits of the interpolation weight as shown in Fig. 3. We originally tried a different strategy of adding the index (an integer) to the interpolation weight (in $[0, 1]$). However, modern processors have a 4 cycle latency when converting an int to a double to execute that addition, and another 4 cycles to extract the index back to an int. Our proposal in Fig. 3 is faster, because it only requires a bit mask and an OR to encode the index (2 cycles), and only a bit mask to decode the index (1 cycle). The precision loss of these two approaches is similar: our proposal loses $log_2(N)$ low-order bits from the interpolation weight, while the alternate approach loses $\lceil log_2(i) \rceil$ bits to encode index $i$. This proposal uses half the memory as the original pair of values, even though we only remove an int, because standard compilers like *Clang* add padding for memory alignment (Raymond, 2019).

Consider an index-weight pair $\langle j, w_j(x) \rangle$, where $w_j(x) \in [0, 1)$ is computed with absolute precision $\pm \epsilon$. Then this encoding method results in a new precision of $\pm(\epsilon \times 2^{\lceil \log_2(N) \rceil})$. The interpolation can then be expressed as $v_j w_j(x) \times (1 \pm (\epsilon \times 2^{\lceil \log_2(N) \rceil})) + v_{j+1}(1 - w_j(x)(1 \pm (\epsilon \times 2^{\lceil \log_2(N) \rceil})))$, which produces an absolute error of at most $2 \max(v_j, v_{j+1}) \times (\pm(\epsilon \times 2^{\lceil \log_2(N) \rceil})$. Note that $\epsilon \approx 2^{-23}$ for single-precision floating point numbers, so the overall error is small, but caution should nevertheless be exercised when constructing deep networks using single precision.

On the real-world ensembles of calibrated lattices models that we tested this on, this technique provided a $5 - 10\%$ speedup when compared against the alternate packing scheme, with larger speed-ups for models with more shared indexing.

## 5 How To Make Multilinear Interpolation Run Faster

Gupta et al. (2016) give an $O(2^D)$ dynamic programming algorithm for multilinear interpolation that iterates over the $D$ inputs, and on each iteration doubles the number of computed weights. Thus the last iteration is half of the work, but for that last iteration, we note that one can interleave in the next step of computing the inner product between the interpolation weights and the LUT values. This interleaving of operations helps the processor be productive while the next value is fetched. This trick is an example of *latency hiding*, a popular computing technique for performing useful work while waiting on a data fetch (Manjikian, 1997; Mowry, 1991), and is critical for performance using accelerators (Chen et al., 2018). To get this latency hiding, we take the trained model parameters, and generate C++ code for the multilinear interpolation such that the compiler will do this interleaving. This provides around a 10-15% end-to-end speed-up for many models.

## 6 How to Make Simplex Interpolation Run Faster

As described in Algorithm 2 of Gupta et al. (2016), the simplex interpolation algorithm requires a sorting permutation $\pi$ over the inputs. On a $D$-dimensional hypercube, the simplex interpolation

requires $D+1$ weights, but constructing the sorting permutation requires $O(D \log(D))$ comparisons which we found dominates the overall runtime.

As with piecewise linear functions, the cost of even a single branch misprediction is high, so we desire a branch-free algorithm. We use *sorting networks* (Batcher, 1968), which construct sequences of max and min operations in order to construct a branch-free sorting implementation for inputs of fixed size.

A sorting permutation would typically be constructed by sorting $\langle key, index \rangle$ pairs, but such paired min-max operations require a comparison followed by six conditional moves. We note that sorting over basic datatypes is much more efficient: min-max on basic datatypes only requires a min and max operation, requiring fewer than half the cycles. This is particularly important for the small sorting problems that arise in simplex interpolation ($D$ is generally $2 - was 25$), since small sorting problems can be handled almost entirely in registers, meaning optimal sorting code is effectively purely computation. To leverage this more efficient sorting, we adopt the same bit-packing technique described in Fig. 3, encoding the index in the low-order bits. The lost precision on the key is $\lceil \log_2(D) \rceil$ bits; a lattice with $2^{32}$ parameters would lose 5 bits of precision. By definition the interpolation weights are each $w_i \in [0, 1)$, so we can bound the absolute error on each $w_i$ by $\epsilon \lceil \log_2(D) \rceil$. As a result, the interpolation output has relative error of at most $\epsilon \lceil \log_2(D) \rceil$.

## 7 PERFORMANCE EVALUATION

To illustrate the overall value of these proposals, some of which synergize, we compare runtimes on four machine-learned multi-layer lattice models: one benchmark and three proprietary (described below).

Our baseline is a C++ interpreter implementation of the interpolation algorithms described in Gupta et al. (2016), with the added speed-up that it uses a fixed auxiliary index-map for each PWL with no transform $T$, $B = 50$ uniform buckets, $\alpha = k_1$, $\beta = (k_n - k_1)/50$. We benchmarked this C++ interpreter code at $397\times$ faster than TensorFlow Lattice (Google AI Blog, 2017) for single-evaluations on a two-layer calibrated lattice model with 4 inputs (the model was just 4 PWLs followed by a four-dimensional lattice interpolated with multilinear interpolation). TensorFlow does get more efficient when evaluated on batches: for a batch size of 4,000 examples, the amortized runtime is only $13\times$ slower than our C++ interpreter baseline.

**Simplex Interpolation:** Fig. 4 shows runtime results for two models where the lattices are interpolated with simplex interpolation. For these we show the baseline interpreter runtime, and the runtime produced by an interpreter with our proposed simplex kernel that uses bit-packing, and a compiled implementation with all of our proposals (faster PWLs and faster simplex) for both double and float. The proposed bit-packing for the simplex kernel does lose a small amount of precision. The worst observed deviation in model output when compared to the C++ implementation was $10^{-13}$ for double-precision evaluation and $10^{-4}$ for single-precision.

The left results in Fig. 4 are for the Kaggle *Wine* dataset using the Kaggle notebook used in Gupta et al. (2018). There are 150 inputs, but all of them are Boolean features except for one continuous feature, which passes through five 40-piece PWLs. The second-layer is an ensemble of 50 lattices, each of which acts on 8 first-layer outputs. Runtime was compared on 84.6k IID examples, and the proposals delivered speed-ups of $5.8 - 9.5\times$.

The right results in Fig. 4 are for a proprietary *selector* model that predicts whether a certain database should be queried for results in response to a given query. The model is an ensemble of 200 calibrated lattices, each on 8 features, so each of the 30 inputs is mapped through an average of 53.33 PWLs in the model's first layer. The PWLs have an average of 15 pieces each. Runtime was compared on 650k IID examples, and the proposals delivered speed-ups of $5 - 6.5\times$.

**Multilinear Interpolation:** Fig. 5 shows runtime results for two models where the lattices are interpolated with multilinear interpolation. For these, we show the baseline interpreter runtime, and the runtime for a compiled implementation with all of our proposals (faster PWLs and latency hiding for multilinear) for both double and float.

The left results in Fig. 5 are for a proprietary model that predicts how long it will take a car to travel a given road. The model is a 4-layer model on 39 inputs, where the first layer passes the 39 inputs

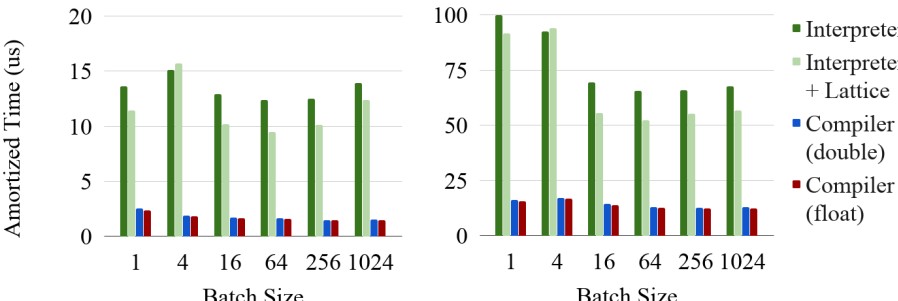

Figure 4: **Left:** Wine Model. The proposed strategies with float (in red) is $5.8 - 9.5\times$ faster than the baseline interpreter (in dark green). **Right:** Selector Model. The proposed strategies with float (in red) is $5 - 6.5\times$ faster than the baseline interpreter (in dark green).

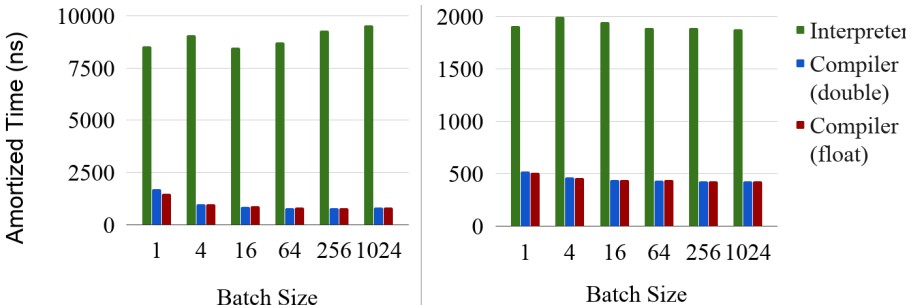

Figure 5: **Left:** Travel Time Estimation Model. The proposed strategies with float (in red) is $5.25 - 11.5\times$ faster than the baseline interpreter (in dark green). The big speed-up here is because there are a lot of PWLs compared to the other operations, and the proposed indexed calibrator strategy takes advantage of the fact that there are 4 PWLs for each of 39 inputs. **Right**: Whole Path Model. The proposed strategies with float (in red) is $3.7 - 4.4\times$ faster than the baseline interpreter (in dark green). For this relatively small model, much of the speed-up came just from moving to the compiler.

through 156 PWLs (each input goes through 4 different PWLs) where each PWL has 50 pieces. The second layer is a linear embedding that maps the 156 calibrated inputs down to four dimensions, then another calibrator layer of PWLs. The final layer takes those four inputs and fuses in another four inputs with an eight-dimensional lattice and multilinear interpolation. Runtime was compared on 94k IID examples, and the proposals delivered speed-ups of $5.25 - 11.5\times$.

The right results in Fig. 5 are for a proprietary model that fuses travel time estimates for different parts of a route into a travel time estimate for the whole path. This model is a 2-layer calibrated lattice model on 8 inputs, with PWLs that have 100–168 pieces. Runtime was compared on 4 million IID examples, and the proposals delivered speed-ups of $3.7 - 4.4\times$.

## 8 CONCLUSIONS AND OPEN QUESTIONS

This paper presents a set of techniques for fast implementations of linear interpolation, using both operation-level optimizations and compiler transformations, which together demonstrate $3 - 11\times$ speed-ups compared to an interpreter-based implementation on several benchmark and real-world models. These speedups are achieved by reducing both fixed overhead costs as well as improving the efficiency of per-example computations. In real-world applications, latency-sensitive ML models must often score one example at a time, so the fixed overheads cannot be reduced by simply batching over examples. Here we focused on CPUs, which are cheap and readily available. Faster solutions may be possible with GPUs, but if the inputs are not already on the GPU, then this may not be a net win due to the kernel launch latency of a GPU. We hypothesize that significantly faster speeds might be possible with FPGAs.

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
