# OpenReview forum: "Fast Linear Interpolation for Piecewise-Linear Functions, GAMs, and Deep Lattice Networks"
_ICLR.cc/2020/Conference — Reject_

### Official Review · AnonReviewer3 · 2019-10-22
**Official Blind Review #3**

**Rating:** 3

**Review:**

The paper proposed several adjustments to speed up the existing implementations of piece-wise linear functions based on Look-up tables (LUT), including an auxiliary LUT through optimisation and sacrificing negligible precision in the standard IEEE754 protocol to reduce the time complexity.

Given my limited experience in the field, I have several concerns and questions after reading this paper several times, which eventually leads to my decision (hopefully it was an educated one):

(1) The paper assumes that the input distribution is known to the system, but it is only true in the testing mode where we generally assume that samples in the test set come from the same distribution which is used to generate training examples. In the case where training is required or finetuning is needed to adapt the system to another distribution, the the proposed auxiliary LUT could have a hard time mapping input data properly.

(2) In my understanding, the transformation, T: supp(P) -> R, is assumed to be known whilst IMO it is crucial to the proposed method. A non-trivial effort is expected in finding T, and a whole area of research is dedicated to this which is Optimal Transport. Currently, plausible and computationally efficient method is through Sinkhorn algorithm. I don't think the paper should downplay the part of finding T as it is crucial.

(3) In the shared Index PWLs, the paper seems to have ignored the fact that different functions still have different output values, and different key-value pairs are indeed needed. For example, log(x) and x^2 (as used in the paper) have different input domains and output domains, so sharing indices between two functions is impossible. Could the authors elaborate on why it is okay to share indices?

(4) A necessary process of LUT from piece-wise linear functions of approximating the ones that we care about is to conduct 1-D density estimation on the input domain, since more fine-grained splits are required in the dense area and less splits elsewhere. The proposed methods in this paper seem to have largely ignored this effect. For example, log(x) needs splits with smaller intervals when the x is around 1.

(5) The presented results only include the speed of the forward computation of several models. It is okay if there is a performance (accuracy or likelihood) and speed trade-off, but the paper didn't present any results on this. Could the authors present some results for this?



**Experience Assessment:**

I do not know much about this area.

**Review Assessment: Checking Correctness Of Derivations And Theory:**

I carefully checked the derivations and theory.

**Review Assessment: Checking Correctness Of Experiments:**

I carefully checked the experiments.

**Review Assessment: Thoroughness In Paper Reading:**

I read the paper thoroughly.

---

> ### Author Response · Authors · 2019-11-11
> **Response to Reviewer 3 Comments**
>
> We hope the following clarifies that the concerns you raise are addressed by the paper.
>
> Response to (1) and (2):
> The only distribution dependence is on the selection of T. Note that in Equation 2 and Equation 3, we propose an objective to optimize to find T, which as the reviewer notes can be expressed as an optimal transport problem. However, because in these applications T must be fast to evaluate, as measured by $C_T$ in equation 3, we can restrict our attention to a small set of possible T that take few instructions. As noted in Section 4.1, we only allow the compiler to select between 3 choices of T in all our experiments (Identity, Log2, Exp -  note that Log2 and Exp are implemented with fast bitwise approximations). While this may not minimize Equation 3, we find empirically that this produces a function T requiring an adjustment cost of at most 3 steps forward. As a result, we do not believe there is significant headroom to improve the choice of T, as more complex choices of T will incur a larger runtime cost $C_T$. As per Equation 3, T is selected based on worst-case behavior over the support of P, and therefore the performance of the algorithm is invariant with respect to distribution shifts within the same support. Additionally, inputs outside of the support of P are clipped to the support and thus do not produce decreases in performance.
>
> Response to (3):
> The shared indices apply to the standard case where the N input keypoints are assigned to be the N quantiles over the input distribution of a particular feature $x$. Note that input distribution is the same for all calibrators acting on the same feature $x$, thus the quantiles are the same, and thus the keys for each of the calibrators are the same and can be shared. Of course, as the reviewer notes, the output values in the key-value pairs are different for each of the calibrators.  (Assigning the N keypoints to the quantiles is done to make sure an equal amount of training data is available to train each output value, to reduce overfitting or degeneracy).
>
> Response to (4):
> For all our experiments, we use the standard quantile initialization of the keys, so they are not trained, and are denser where the feature $x$ is denser. A more accurate piecewise linear function is obtained by using more keypoints K. As we have shown in this paper, our runtime is largely independent of K as long as you compile for that K, though there can be nonlinear caching impact of increasing K.
>
> Response to (5):
> As noted in Section 7, the output values differed by no more than $10^{-13}$ for a double implementation, but did differ by up to $10^{-4}$ for a float implementation. In a final version, we will add the effect on classifier accuracy of these small changes.

---

### Official Review · AnonReviewer4 · 2019-10-23
**Official Blind Review #4**

**Rating:** 3

**Review:**

This paper proposes several low-level code optimizations aimed at speeding up evaluation time for linearly interpolated look-up tables, a method often used in situations where fast evaluation times are required. The paper is very practically oriented in the sense that it does not aim for improving asymptotic speed-ups, but rather real speed-ups expressible in terms of CPU cycles that are achievable by exploiting compiler optimizations such as branch prediction and loop unrolling. The focus is on unbatched computations which can arise in many real-time situations. The proposed implementation techniques are as follows:
- A method for fast index mapping, which first transforms inputs using a monotonic function such as log_2 or 2^x, and the applying a branch-free linear search implementation on a uniformly-spaced auxiliary LUT.
- A memory-efficient bit-packing technique to store both integer and floating point representations together.
- Speed-up for multilinear interpolation using latency hiding
- Branch-free implementation of sorting permutation, needed for simplex interpolation
The proposed implementation is then evaluated on 4 different benchmarks, with considerable speed gains over interpreter-based baselines, while batching and single-vs-double precision do not have major impact on speed.

While the paper is relevant and interesting, and the proposed techniques are reasonable and probably result of a considerable amount of work, more effort is needed to improve clarity and preciseness of the explanations, and (most importantly) the experimental evaluation. Detailed strengths and weaknesses are outlined below.

Strengths
- The paper is well-motivated and relevant to the ML community
- Low-level speed optimizations are needed but overlooked in the community
- Reasonable choice of experimental conditions (focus on unbatched CPU evaluation, testing on a selection of 4 different tasks)
- Proposed techniques are sensible

Weaknesses (roughly in order of decreasing significance)
- Gains over Tensorflow performance is advertised in the intro, but only mentioned anecdotally in the experiments. Also, the Tensorflow implementation should be briefly explained to make clear where these gains come from.
- The experiments put much focus on speed performance over different batch sizes, but this is (1) not the focus of the paper (unbatched CPU operation is the focus), and (2) is little informative because the introduced methods (which do not benefit much from batching) are not compared against Tensorflow (which does benefit from batching).
- No ablation study is presented. The method description mentions informally how much speed-up is to be expected from different parts of the proposed method, but do not clarify how much this contributes to overall speed-gains.
- The description in 4.1 is rather hard to follow, even though the ideas behind it are relatively simple. An illustrative figure might be of help for readers.
- The method description in 4.1 lacks formal preciseness. For example, in 4.1. alpha, P, supp, are not defined (or in some cases introduced much after they are used first), and the “+4” in the beginning of page 5 appears out of nowhere.
- The proposed bit-packing is not well motivated. It promises to save half of the memory at a minor loss in precision (at least for the double-precision case), but it is unclear how much of a bottleneck this memory consumption is in the first place. In addition, it remains unclear to me why this is relevant particularly in the shared index case.
- While the topic is relevant for this conference, readers are likely not familiar with some of the concepts used in the paper, and a bit more explanation is needed. An example for this is “branch prediction”, which is a key concept; readers unfamiliar with this compiler concept will likely not understand the paper, and a brief explanation is needed. Another example is “loop-carry dependency”, a term that could be explained in a short footnote. A third example is FPGA, which is mentioned in the very last sentence without further explanation/justification.
- The introduction could be a bit more concrete on describing tasks where PWLs are relevant

**Experience Assessment:**

I do not know much about this area.

**Review Assessment: Checking Correctness Of Derivations And Theory:**

I assessed the sensibility of the derivations and theory.

**Review Assessment: Checking Correctness Of Experiments:**

I carefully checked the experiments.

**Review Assessment: Thoroughness In Paper Reading:**

I read the paper thoroughly.

---

> ### Author Response · Authors · 2019-11-12
> **Response to Reviewer 4 Comments**
>
> We agree with the assessment of the paper’s strengths, and have some feedback on the noted weaknesses.
>
> Re: “Gains over Tensorflow performance is advertised in the intro, but only mentioned anecdotally in the experiments. Also, the Tensorflow implementation should be briefly explained to make clear where these gains come from.”
> We will include more experimental results against Google’s TensorFlow implementation for linear interpolation models (TensorFlow Lattice). In general, TF is slow due to dispatch time, and generally doesn’t handle operations of sparse sets of data efficiently, and TF is especially ill-suited for simplex interpolation because of its sort operation.
>
> Re: “The experiments put much focus on speed performance over different batch sizes, but this is (1) not the focus of the paper (unbatched CPU operation is the focus), and (2) is little informative because the introduced methods (which do not benefit much from batching) are not compared against Tensorflow (which does benefit from batching).”
> We note that even at large batch sizes, the Tensorflow implementation is substantially slower than the interpreter on models using kernels well-suited for a Tensorflow implementation.
>
> Re: “No ablation study is presented. The method description mentions informally how much speed-up is to be expected from different parts of the proposed method, but do not clarify how much this contributes to overall speed-gains.”
> We will add ablation studies (Fig. 2 does show the effect of just the optimized index-mapping).
>
> Re: “The description in 4.1 is rather hard to follow, even though the ideas behind it are relatively simple. An illustrative figure might be of help for readers.”
> Yes, we will include a illustrative figure in a final version.
>
> Re: “The method description in 4.1 lacks formal preciseness. For example, in 4.1. alpha, P, supp, are not defined (or in some cases introduced much after they are used first), and the “+4” in the beginning of page 5 appears out of nowhere.”
>
> Thanks for the note, we will make these definitions clearer. The “+4” figure comes from a calculation based on L1 cache access latency and arithmetic operations as given by Fog 2018, and was noted in section 2.1, but we’ll make it clearer why it’s showing up in Sec 4.1
>
> Re: “The proposed bit-packing is not well motivated. It promises to save half of the memory at a minor loss in precision (at least for the double-precision case), but it is unclear how much of a bottleneck this memory consumption is in the first place. In addition, it remains unclear to me why this is relevant particularly in the shared index case.”
>
> The bit-packing is only used in the shared-index case, and not the single-calibration case. We note that memory latency is important, and due to the large numbers of inputs in our models, this optimization increases the size of models which can be executed using stack-allocated memory instead of heap-allocated memory, reducing the number of much costlier cache misses. Note that this also affects the batch size that can be executed on stack before spilling into heap memory.
>
> Re: “readers are likely not familiar with some of the concepts used in the paper, and a bit more explanation is needed.”
> We will add brief explanations of these terms.
>
> Re: The introduction could be a bit more concrete on describing tasks where PWLs are relevant.
> Without the restriction of blind review, we will be able to give additional concrete examples.  GAMs (which either directly use a sum of PWLs or can be approximately implemented with PWLs) are a standard modeling approach by statisticians, and many papers can be found applying GAMs to real-world problems.  Here’s a nice recent public post by Microsoft on the importance of GAMs: https://docs.microsoft.com/en-us/dotnet/machine-learning/how-to-guides/use-gams-for-model-explainability
> See also this Facebook paper on using GAMs for forecasting:
> https://peerj.com/preprints/3190/

---

### Official Review · AnonReviewer1 · 2019-10-26
**Official Blind Review #1**

**Rating:** 1

**Review:**

This paper proposes two techniques for fast linear interpolation on CPUs. They achieved speedups by reducing 1) fixed overhead cost and 2) per example computation (linear interpolation operation level optimization).
Authors consider this problem for small operation models like linear interpolation rather than the models requiring large operations such as ResNet. In this case, dispatch overhead cannot be ignored and so they use the MLIR frameworks to optimize trained model into the C++ code (reducing fixed overhead cost). This results in 2-3x speed up. Secondly, they propose the way to construct auxiliary index-mapping function by considering spacing of the key points rather just using for example evenly spaced index-mapping.
They compare proposed method to C++ interpreter implementation on two-layer and deep lattice networks and achieve 5-10x speed improvements.

It seems the topic of this paper does not fit ICLR and most machine learning researchers are unlikely to be interested in and even understand this paper. This reviewer also does not have enough knowledge and background to judge this paper. But my impression is that achieving speed up using existing MLIR framework has no surprising novelty.
Moreover, the experimental results seems quite limited in the sense that they only experiment with trained 2 and 4-layer calibrated lattice models which are kind of small.

It would be better to highlight why the proposed method is meaningful and provide more background knowledge to understand this paper.

This is only consider optimization on CPUs. What about the case of using GPUs?

Is branch free assumption for functions ‘Adjust’ & ‘T’ is valid? (I don’t have much knowledge on compiler..)

**Experience Assessment:**

I do not know much about this area.

**Review Assessment: Checking Correctness Of Derivations And Theory:**

I did not assess the derivations or theory.

**Review Assessment: Checking Correctness Of Experiments:**

I assessed the sensibility of the experiments.

**Review Assessment: Thoroughness In Paper Reading:**

I made a quick assessment of this paper.

---

> ### Author Response · Authors · 2019-11-12
> **Response to Reviewer #1.**
>
> We hope these responses give you more perspective on the novelty and significance.
>
> Re: “But my impression is that achieving speed up using existing MLIR framework has no surprising novelty. “
> The key novelty here is that we show that ML models with many small ops (here, linear interpolation ops) are not bottlenecked by their computation, but by the interpreter itself, and we additionally present low-level optimizations and data-handling that enable incredibly fast runtimes on widely available CPUs.  This is in contrast to ML models with large matrix operations (like ResNet or DNN’s) where the interpreter is a small fraction of total runtime, and have well-studied primitives.
>
>
> Re: “This is only consider optimization on CPUs. What about the case of using GPUs?”
> Evaluation runtimes in our experiments are at or below the typical GPU latency of around 10 microseconds, so our CPU implementation would already have finished by the time you moved the data to a GPU.
>
> Re: “Is branch free assumption for functions ‘Adjust’ & ‘T’ is valid? (I don’t have much knowledge on compiler..)”
> Yes, please see Section 2.1 for details.
>
>
> Re: “trained 2 and 4-layer calibrated lattice models which are kind of small.”
> Yes, the models in the experiments ranged from very small to medium-sized, the largest was the selector model that had 75,000 parameters.  This paper is focused on models with runtimes in the microseconds or nanoseconds.  In production systems such models can run early in a latency-critical pipeline, or many such models may need to be run serially, making speed-ups important.

---

### Decision · Program_Chairs · 2019-12-19

**Decision:**

Reject

**Comment:**

This paper proposes an efficient implementation of piecewise linear functions.

While this paper tackles a problem of large apparent interest, as noted by the reviewers the paper (1) is pretty far from the domain of the average ICLR paper, and (2) not written with the high standards of clarity that would make it accessible to the average ICLR reader. I am not impugning on the merits of the paper itself, but would suggest that the authors both take the reviewer's advice with regards to the clarity issues (among other) and consider submitting to the Systems for ML workshop, a systems conference, a compilers conference, or some other venue with a larger percentage of qualified readers (and reviewers).